# On the Potential Role of the (Pseudo-) Jahn–Teller Effect in the Membrane Transport Processes: Enniatin B and Beauvericin

**DOI:** 10.3390/molecules28176264

**Published:** 2023-08-27

**Authors:** Dagmar Štellerová, Vladimír Lukeš, Martin Breza

**Affiliations:** Institute of Physical Chemistry and Chemical Physics, Slovak University of Technology in Bratislava (STU), Radlinskeho 9, SK-81237 Bratislava, Slovakia; dagmar.stellerova@stuba.sk (D.Š.); vladimir.lukes@stuba.sk (V.L.)

**Keywords:** DFT, natural compounds, tautomers, symmetry, mycotoxin

## Abstract

The molecular structure of mycotoxins enniatin B and beauvericin, which are used as ionophores, was studied using density functional theory in various symmetry groups and singly charged states. We have shown that the charge addition or removal causes significant structural changes. Unlike the neutral C_3_ molecules, the stability of the charged C_1_ structures was explained by the Jahn–Teller or Pseudo-Jahn–Teller effect. This finding agrees with the available experimental X-ray structures of their metal complexes where electron density transfer from the metal can be expected. Hence, the membrane permeability of metal sandwich-structure complexes possessing antimicrobial activities is modulated by the conformational changes.

## 1. Introduction

Symmetry, a common feature found in nature and chemistry, seems to bypass compounds produced by living systems. In general, high-symmetric organic molecules are toxic and highly carcinogenic. However, there are few types of biomolecules in plants and fungi that possess symmetry, such as the plant pigment curcumine (diferuloylmethane) with antioxidant, antibacterial, anti-inflammatory, and antitumor activity [1,2]. Among the small number of symmetric molecules produced by fungi, those of the *Fusarium* genus attract extensive attention due to their frequent contamination of food commodities, e.g., cereal grains [3,4]. They produce a number of secondary metabolites, which are questioned due to their potential health hazards [5,6]. Among them, enniatin B (**ENB**) and beauvericin (**BEA**) belong to the most studied mycotoxins due to their prevalent occurrence in cereals [7,8,9]. Although toxic, they have shown numerous biological activities, including antibiotic, antimicrobial, insecticidal, anthelmintic, anticancer and apoptosis-inducing effects [10,11,12,13,14,15,16,17,18,19,20].

Their molecular structure consists of three D-hydroxyisovaleric acids in alternation with three N-methyl branched-chain L-amino acids. **ENB** contains three N-methyl valine residues, whereas in **BEA**, there are three N-methyl phenylalanine moieties (see Figure 1). The interior of their structures contains strategically spaced oxygens with free electron pairs capable of forming ion–dipole interactions with mono- and divalent cations. The lipophilic exterior of their molecules enables them to incorporate into biological membranes and thus create cation-selective pores. As a result, disturbances of the action potential, metabolic state, and cell homeostasis that lead to cell death may occur [21,22]. Due to the size of small depsipeptide ring sizes, their six carbonyl oxygen atoms must be alternately arranged above and below the mean molecular plane, while the peptide and ester groups are in *trans*-positions and approximately planar [23]. Further studies revealed that the biological action of these compounds is related to their ability to induce transmembrane cation transport, i.e., act as ionophores [24,25]. Related experiments showed [26] that the membrane effect is associated with their ability to form stable complexes with the cations of alkaline and alkaline-earth metals. As experimentally shown by Ovchinnikov et al. [26], the complexes formed undergo conformational changes depending on the solvent used and on the central atom metal. The flexible depsipeptide chain of enniatins can be adjusted to the size of the metal ion and causes the low selectivity of these complexones. The 2:1 and 3:2 complexes are formed where the sandwiched cations on the symmetry axis are coordinated prevailingly by the carbonyl oxygen atoms. It implies better screening of the metal cation from the anion and solvent in comparison with the 1:1 complexes and higher solubility in organic solvents.

The schematic structure of **ENB** and **BEA** depicted in Figure 1 reveals that the highest possible symmetry point group of their molecules is C_3_. This fact opens the question of whether the Jahn–Teller (JT) or Pseudo-Jahn–Teller (PJT) effect can particularly modulate the energy levels of these symmetric compounds, and which consequences can be awaited from this effect. For the symmetric neutral enol and keto forms of curcumine, we have recently demonstrated the potential significance of the PJT effect [27], which may rationalize the photoprotection action and activity of naturally occurring symmetric dyes in plants.

Although many published works deal with the X-ray structure of naturally occurring **ENB** and **BEA**, which are collected in the Cambridge Structural Database (CSD), analysis of their symmetry space group is not available. Therefore, the partial aims of this study are: (1) to analyze the symmetry of **ENB** and **BEA** structures available in CSD; (2) to perform optimization of the gas-phase structures in various symmetric groups and charged states using Density functional theory; and (3) to explain the stable structures of lower symmetry using (P)JT treatment. Finally, the possible (P)JT effect for the investigated molecules will be estimated.

## 2. Theoretical Background

The JT theorem [28] states that any non-linear configuration of atomic nuclei in a degenerate electron state is unstable. Therefore, at least one stable nuclear configuration of lower symmetry must be obtained during a symmetry decrease where the electron degeneracy is removed. The adiabatic potential surfaces (APSs) of such systems can be described by an analytic function based on perturbation theory. Its minimization produces the atomic coordinates of the corresponding stable systems [29]. For large systems, this treatment is too complicated. This problem can be simplified using group theory, and the symmetry of the stable structures can be predicted.

In the method of step-by-step symmetry descent [30,31] the removal of some symmetry elements from the parent ‘unstable’ high-symmetry group (i.e., symmetry decrease) with a multidimensional irreducible representation, which corresponds to a molecule in a degenerate electron state, causes consecutive splitting of this representation into its ‘stable’ subgroups until a nondegenerate electron state (i.e., one-dimensional representation) is obtained. In our case, molecules of the C_3_ symmetry group in a double-degenerate E electron state undergo a symmetry descent to the C_1_ symmetry (the C_3_ symmetry axis is removed).

Alternatively, the epikernel-principle method [32] is based on the JT active distortion coordinate *Q* of Λ representation for a degenerate electron state Ψ of Γ representation within the parent symmetry group. Λ is the non-totally symmetric part of the symmetrized direct product [Γ ⊗ Γ], which corresponds to a non-vanishing value of $<\Psi \left|\frac{\partial^{\;}\hat{H}}{\partial Q}\right|\Psi >$ integrals where $\hat{H}$ denotes Hamiltonian. For the molecules of the C_3_ symmetry group in a double-degenerate E electron state:[E ⊗ E] = A ⊕ E(1)

Here, we obtain the JT active coordinate of the E representation. According to the epikernel principle, the extrema of a JT energy surface correspond to the kernel K(G, Λ) or epikernel E(G, Λ) subgroups of the parent group G. Kernels contain symmetry operations that leave the Λ representation invariant, whereas epikernels leave invariant-only some components of the degenerate Λ representation. In our case,
K(C_3_, E) = E(C_3_, E) = C_1_(2)
is in agreement with the above-mentioned method of step-by-step symmetry descent. The energy difference between the high-symmetry unstable and low-symmetry stable structures of the same compound is denoted as the Jahn–Teller stabilization energy *E*_JT_.

A similar instability of a high-symmetry structure, known as the PJT effect [33], may be observed in the case of sufficiently strong vibronic coupling between two pseudodegenerate electronic states (usually ground and excited). Using perturbation theory, the energy surface for the simplest case of two interacting electronic states Ψ_1_ and Ψ_2_ of different space symmetries can be described by the formula
(3)$$E\left(Q\right)=\frac{1}{2}KQ^{2}\pm [\frac{{∆}^{2}}{4}+F^{2}Q^{2}{]}^{1/2}$$
where *E* is the energy of the electronic state, *Q* is the distortion coordinate, Δ is the difference in energy between both electronic states in the undistorted geometry, *K* is the primary force constant (without vibronic coupling) and *F* is the vibronic coupling constant. The curvature of the lower state is negative if the energy difference is
(4)$$∆<2\frac{F^{2}}{K}$$

Therefore, an instability in the *Q* direction can be concluded. Otherwise, the stable structure corresponds to *Q* = 0, that is, the high-symmetry structure is preserved despite the diminished curvature of the lower energy state. Higher Δ values imply a weaker PJT interaction. Vibronic interactions between the states of various spin multiplicities are forbidden because their spin states are orthogonal. In general, PJT interactions are possible in any system. Only if their consequences are observed (in spectra, structure, reactivity, etc.) can the corresponding vibronic interaction be denoted as the PJT effect.

The symmetry of stable PJT structures can be predicted using the epikernel principle method [34,35]. For non-vanishing values of $<\Psi_{1}\left|\frac{\partial^{\;}\hat{H}}{\partial Q}\right|\Psi_{2}>$ integrals, the representation Λ of the JT active coordinate *Q* is corresponding to the non-totally symmetric part of the direct product of Γ_1_ and Γ_2_ representations of electronic states Ψ_1_ and Ψ_2_, respectively. The symmetry groups of stable structures correspond to the kernel K(G, Λ) or the epikernel E(G, Λ) subgroups of the parent group G (see above). For the molecules of the C_3_ symmetry group in a non-degenerate A ground electron state, excited E states can produce non symmetric JT active coordinates because of the direct product
A ⊗ E = E(5)

According to Equation (2), we again obtain stable structures of the C_1_ symmetry group.

## 3. Results

In the Cambridge Structural Database (CSD) [36], we have found five X-ray structures that contain a neutral **ENB** molecule (Table 1). Moreover, three additional X-ray structures of enniatin B complexes with alkaline metals were acquired as well. We checked the molecular symmetry using the distance between neighboring carbonyl oxygens O_C=O_ (which might be bonded to metal cations), as well as bridging oxygens O_bridge_ and nitrogens N and confirmed the results by the (O–O–O)_C=O_, (O–O–O)_bridge_ and N–N–N angles between the atoms mentioned (Figure 2). Despite too high R-factors (and missing standard deviations) of some structures, we confirmed the C_3_ symmetry axis in neutral **ENB** molecules in the BICMEF, DESYIJ, and EROPIM structures, as well as in its potassium complex in the IHECUT structure. Fewer results were obtained for beauvericin. Only one X-ray structure containing its neutral molecule and a single beauvericin complex with barium was found in CSD (Table 1). Using the treatment mentioned above, there is no C_3_ symmetry axis in these structures (Table 2). This indicates the symmetry descent, which may be ascribed to vibronic interactions, i.e., some sort of (P)JT effect.

In the first step of our model study, we tried to optimize all systems within the C_3_ symmetry group. However, we have not found any C_3_ structures of cationic **ENB** and charged **BEA** species because they adopt a double-degenerate ground electron state that is not accessible within DFT treatment. Therefore, the JT effect reduces their molecular structures to C_1_ symmetry, as illustrated in Table 3. Neutral **ENB** and **BEA** molecules are the most stable, whereas their cations are the least stable. The calculated energies of their anions are close to those of the neutral ones. This reflects their affinity to form metal complexes (see Table 2). We have found anionic **ENB** structures of C_3_ symmetry, which are unstable (imaginary vibration of e representation), and the stable ones of C_1_ symmetry. Their existence might be explained by the PJT effect (see below).

The data collected in Appendix A reveal that the electronic excited states of the systems under study are too high. The lowest energy vertical electron energy transition (S_0_→S_1_) is 5.368 eV (231 nm) for **ENB** and 5.363 eV (231 nm) for **BEA**. These energy values correspond to the UV region in agreement with experimental observations [26,37]. Vibronic interactions with double-degenerate excited states of much lower energies (Appendix A) produce an imaginary vibration (i.e., PJT active coordinate) of e symmetry that leads to the stable structure of the C_1_ symmetry group with *E*_JT_ of 0.61 eV. However, the correlation of excited states of anionic **ENB** between the C_3_ and C_1_ symmetries is problematic. The electronic excitation energies in the stable C_1_ structure are much higher, and the corresponding oscillator strengths are very low. If we assume that small PJT perturbation does not cause significant changes in oscillator strengths, the 1^2^E electron state in the C_3_ structure, with very high oscillator strength, has no counterparts among low excited states in the C_1_ structure. This implies that either this assumption is incorrect (i.e., the PJT effect does not correspond to a small perturbation), or that the 1^2^E electron state does not undergo the above vibronic interaction.

We checked the molecular symmetry of all **ENB** and **BEA** species by interatomic distances and angles between heteroatoms (Table 4), as well as by their natural charges and spin populations (Table 5). The mutual distances between neighboring carbonyl O atoms in neutral molecules are shorter than those in cationic species and longer than those in anionic species. The reverse trend holds for the distances between the remaining heteroatoms. This implies a higher C=O perpendicularity to the main molecular plane in anionic species. A large benzyl substituent in **BEA** repels C=O groups more toward the center of the depsipeptide ring than the isopropyls in **ENB**.

Bridging O atoms in our systems are less negative than the carbonyl ones (Table 5). As expected, the negative charges of heteroatoms are higher in the anionic species, whereas in the cationic ones, they are lower than in the neutral molecules. The N atoms in the cationic C_1_ species have the highest spin populations. The spin density distribution in charged species is highly non-symmetric (Figure 3 and Figure 4). In the beauvericin anion, the spin density is located mainly at aromatic rings, whereas in remaining systems it is prevailingly located in the area of a single N atom and neighboring carbonyl. This asymmetric distribution should increase the reactivity of charged species.

## 4. Method

The geometries of the neutral (charge *q* = 0), cationic (*q* = +1) and anionic (*q* = −1) molecules of **ENB** and **BEA** (Figure 2) in the spin states of the ground singlet (*q* = 0) or doublet (*q* = +1, −1) were optimized within the C_3_ and C_1_ symmetry point groups using the M06-2X hybrid functional [38] (a very recent functional with dispersion corrections), combined with standard cc-pVDZ basis sets for all atoms taken from the Gaussian library (we were not able to use larger basis sets because of technical restrictions). Stability of the obtained geometries was checked on imaginary vibrations by vibrational analysis. The Gibbs free energies were computed for room temperature (298.15 K, see Appendix A for details). Natural charges and natural spin populations on atoms were evaluated using Natural Bond Orbitals (NBO) analysis [39,40], which are more reliable than the analogous results obtained by Mulliken population analysis. Time-dependent DFT (TD-DFT) treatment [41,42] (more frequently used than single-excitation Configuration Interaction methods) for up to 30 vertical electronic states was used for excited state calculations. Gaussian16 (Revision B.01) software [43] was used for all quantum-chemical calculations because it brings more new methods, property predictions and performance enhancements than similar quantum-chemical software. The MOLDRAW (https://www.moldraw.software.informer.com, accessed on 9 September 2019) [44] and Molekel (Version 5.4.0.8) [45] software were used for geometry manipulation and visualization purposes.

## 5. Conclusions

We have shown that the charged species of compounds under study exhibit JT symmetry descent, unlike neutral molecules. It agrees with the experimental X-ray structures of their metal complexes (Table 2), where electron density transfer from the metal can be expected. Analysis of the symmetry changes reveals the existence of three stable C_1_ geometries as a consequence of the JT or PJT effect in their C_3_ parent structures. This fluxional behavior might explain the experimentally observed equilibria between various forms of metal complexes [26]. Our study indicates that the membrane permeability of the sandwich structure complexes mentioned above is modulated by conformational changes between various forms, which can be explained by the (P)JT effect. Further theoretical study of alkaline and alkaline-earth metal complexes is desirable to verify this hypothesis. Our approach can be used for any symmetric configuration of atomic nuclei undergoing the (P)JT effect.

## Figures and Tables

**Figure 1 molecules-28-06264-f001:**
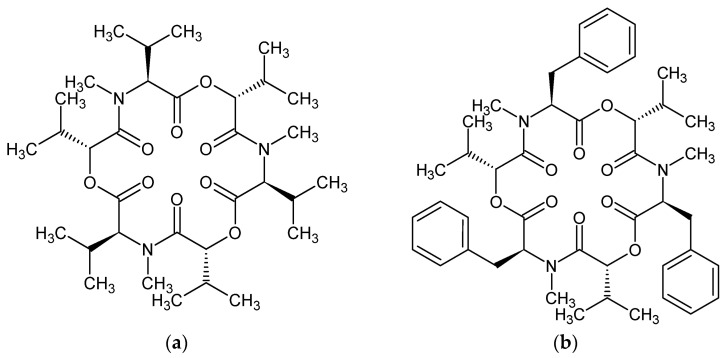
Schematic structure and atom notation of enniatin B (**a**) and beauvericin (**b**).

**Figure 2 molecules-28-06264-f002:**
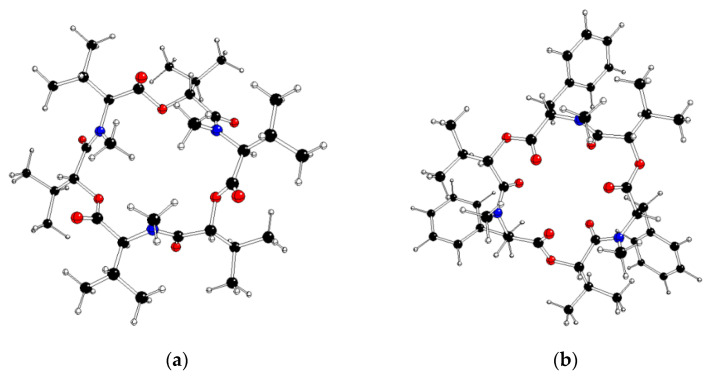
M06-2X/cc-pVDZ optimized gas phase structure in C_3_ symmetry group of **ENB** (**a**) and **BEA** (**b**) (O—red, N—blue, C—black, H—white).

**Figure 3 molecules-28-06264-f003:**
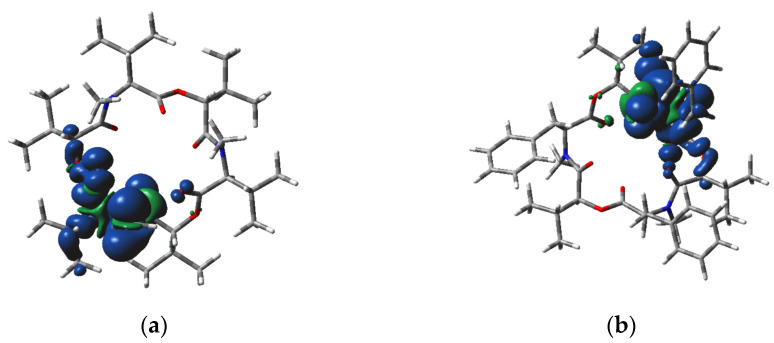
M06-2X/cc-pVDZ calculated spin density distribution over the cation of (**a**) enniatin B and (**b**) beauvericin in C_1_ symmetry (O—red, N—blue, C—grey, H—white, positive, and negative spin density at the 0.0005 a.u. iso-surface is indicated as dark blue and green, respectively).

**Figure 4 molecules-28-06264-f004:**
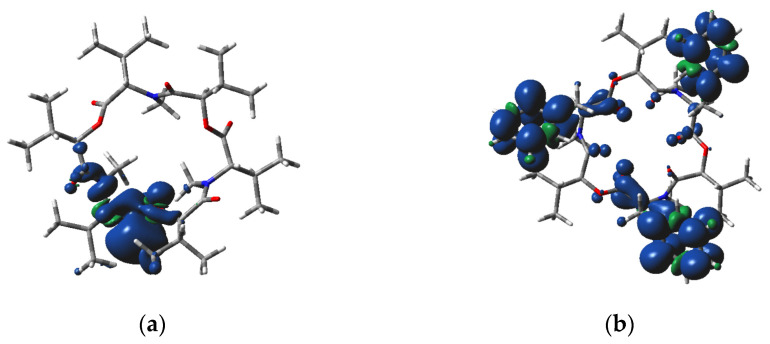
M06-2X/cc-pVDZ calculated spin density distribution over the anion of (**a**) enniatin B and (**b**) beauvericin in C_1_ symmetry (O—red, N—blue, C—grey, H—white, positive, and negative spin density at the 0.0005 a.u. iso-surface is indicated as dark blue and green, respectively).

**Table 1 molecules-28-06264-t001:** X-ray structures of the molecules under study found in CSD [36].

CSD Code	Chemical Formula	Chemical Name	Space Group	R-Factor
Neutral **ENB** molecules				
BICMEF	C_33_H_57_N_3_O_9_·1.5H_2_O	Cyclo-tris(l-methylvalyl-d-2-hydroxyisovaleryl) sesquihydrate	R3	0.06
CIKJAH	C_33_H_57_N_3_O_9_·1.67H_2_O	Enniatin B hydrate	R3	0.057
DESYIJ	C_33_H_57_N_3_O_9_	Enniatin B unknown solvate	R3	0.113
EROPIM	(C_11_H_19_NO_3_)_3_·0.2238(CH_3_O)_3_· 0.328677O	3,6,9,12,15,18-hexaisopropyl-4,10,16-trimethyl-1,7,13-trioxa-4,10,16-triazacyclooctadecane-2,5,8,11,14,17-hexone methanol solvate hydrate	R3	0.0397
ZASQOZ	C_33_H_57_N_3_O_9_	Enniatin B	P1_1_2_1_	0.0714
**ENB** complexes				
IHECUT	(C_132_H_229_K_4_N_12_O_36_)_n_(CNS)_4n_	catena-(tetrakis((μ_2_-Enniatin B)-potassium thiocyanate))	P3	0.0643
MVHIRB10	(C_34_H_57_N_4_O_9_RbS)_n_	catena-(l,d,l,l,d,l-Enniatin B)-rubidium isothiocyanate	P4_3_2_1_2	0.085
PEKFEQ	(C_66_H_114_KN_6_O_18_)I·H_2_O	bis(Enniatin B)-potassium iodide monohydrate	P6_1_	0.0766
Neutral **BEA** molecule				
BEVERC	C_45_H_57_N_3_O_9_·H_2_O	Cyclo-tri(l-*N*-methylphenylalanyl-d-alpha-hydroxyisovaleryl) monohydrate	P2_1_	0.096
**BEA** complex				
BEAVBA	(C_45_H_57_N_3_O_9_)(C_6_H_2_N_3_O_7_)_2_Ba·2C_7_H_8_	Beauvericin barium picrate toluene solvate	P2_1_2_1_2	0.15

**Table 2 molecules-28-06264-t002:** Molecular point groups Γ, neighboring carbonyl O–O distances *d*(O–O)_C=O_, bridging O–O distances *d*(O–O)_bridge_, N–N distances *d*(N–N), neighboring carbonyl O–O–O angles *Θ*(O–O–O)_C=O_, bridging O–O–O angles *Θ*(O–O–O)_bridge_, and N–N–N angles *Θ*(N–N–N) in X-ray structures of **ENB** and **BEA**. All distances are in Å and all angles are in degrees.

System	Γ	*d*(O–O)_C=O_	*d*(O–O)_bridge_	*d*(N–N)	*Θ*(O–O–O)_C=O_	*Θ*(O–O–O)_bridge_	*Θ*(N–N–N)
Neutral **ENB** molecules							
BICMEF	C_3_	4.827(3×)	5.636(3×)	5.497(3×)	74.6(3×)	60.0(3×)	60.0(3×)
		4.725(3×)			71.9(3×)		
CIKJAH	C_1_	4.86(1), 4.70(1)	5.577(9)	5.570(7)	74.7(2), 82.6(2)	60.0(1) (3×)	60.0(1) (3×)
		4.31(1), 3.66(1)	5.58(1)	5.570(9)	74.7(2), 81.6(2)		
		4.86(1), 4.70(1)	5.577(9)	5.57(1)	74.7(2), 73.1(2)		
DESYIJ	C_3_	4.01(3) (3×)	5.87(2) (3×)	5.71(3) (3×)	67.5(5) (3×)	60.0(3) (3×)	60.0(3) (3×)
		3.89(3) (3×)			66.0(5) (3×)		
EROPIM	C_3_	4.888(2) (3×)	5.556(3) (3×)	5.444(3) (3×)	75.07(4) (3×)	60.00(3) (3×)	60.00(4) (3×)
		4.776(3) (3×)			74.57(4) (3×)		
ZASQOZ ^(a)^	C_1_	4.746(9), 3.41(1)	5.829(9)	5.989(8)	82.8(2), 74.2(2)	56.38(9)	52.8(1)
		3.69(1), 4.26(1)	5.90(1)	5.510(9)	57.4(2), 48.2(1)	61.2(1)	68.4(1)
		4.907(9),4.969(9)	5.540(7)	5.13(1)	72.7(2), 83.5(2)	62.4(1)	58.8(1)
		4.92(1), 4.805(9)	5.631(8)	5.455(9)	84.8(2), 74.0(2)	59.8(1)	51.8(1)
		4.10(1), 3.72(1)	5.91(1)	5.97(1)	56.5(2), 73.5(3)	57.7(1)	58.8(1)
		3.64(1), 4.77(1)	5.762(8)	5.01(1)	86.1(3), 68.1(2)	62.5(1)	69.4(1)
**ENB** complexes							
IHECUT ^(b)^	C_3_	3.40(1) (3×)	5.87(2) (3×)	6.07(2) (3×)	68.6(3) (3×)	60.0(2) (12×)	60.0(2)(12×)
		3.37(2) (3×)			64.7(3) (3×)		
		3.42(1) (3×)	6.04(1) (3×)	5.93(2) (3×)	67.4(3) (3×)		
		3.52(2) (3×)			67.3(3) (3×)		
		3.24(1) (3×)	6.27(2) (3×)	6.02(2) (3×)	67.9(4) (3×)		
		3.59(2) (3×)			64.6(4) (3×)		
		3.45(1) (3×)	5.80(2) (3×)	5.86(2) (3×)	66.7(3) (3×)		
		3.46(1) (3×)			63.2(3) (3×)		
MVHIRB10	C_1_	3.26, 3.52	5.85	5.84	54.3, 91.4	56.8	65.5
		3.70, 3.28	5.80	5.46	90.9, 66.8	65.7	58.3
		3.63, 4.24	6.32	5.32	63.3, 123.6	57.5	56.1
PEKFEQ ^(a)^	C_1_	3.64(5), 3.64(4)	5.93(4)	5.93(4)	68.5(7), 65.9(7)	60.8(4)	61.1(5)
		3.55(3), 3.80(3)	6.01(3)	5.82(3)	68.0(7), 62.9(6)	58.9(4)	59.6(5)
		3.67(4), 3.63(4)	5.94(3)	6.05(5)	70.5(7), 62.2(7)	60.3(4)	59.3(5)
		3.62(4), 3.55(4)	5.90(2)	5.99(3)	77.3(9), 61.6(9)	60.8(4)	58.8(4)
		3.69(3), 3.48(3)	6.02(4)	6.20(5)	77.7(9), 64.6(7)	59.5(4)	62.3(5)
		3.76(4), 3.62(3)	5.99(3)	5.98(3)	68.5(7), 56.0(6)	59.7(4)	58.9(4)
Neutral **BEA** molecule							
BEVERC	C_1_	3.21, 4.03	6.24	5.70	52.5, 78.9	58.0	64.2
		3.55, 4.06	6.30	5.94	62.4, 70.7	60.5	56.0
		3.95, 3.49	6.08	6.19	68.5, 73.9	61.6	59.8
BEA complex							
BEAVBA	C_1_	3.33, 3.46	6.33	5.92	67.6, 69.4	61.2	59.6
		3.36, 3.82	6.20	5.90	63.9, 74.2	59.1	60.0
		3.11, 3.49	6.23	5.88	69.2, 66.4	59.7	60.4

Remarks: ^(a)^ two independent **ENB** molecules; ^(b)^ four independent **ENB** molecules.

**Table 3 molecules-28-06264-t003:** Electronic DFT energies (*E*_DFT_), Gibbs free energies at room temperature (*G*_298_), Jahn–Teller stabilization energies (*E*_JT_), and symmetries of imaginary vibrations for optimized geometries of enniatin B and beauvericin in various charge (*q*) and spin states.

Compound	*q*	Symmetry Group	Ground Electron State	*E*_DFT_ [Hartree]	*G*_298_ [Hartree]	*E*_JT_ [eV]	Imaginary Vibration Symmetry
**ENB**	0	C_3_	^1^A	−2132.49018	−2131.68230	0.000	-
	+1	C_1_	^2^A	−2132.19790	−2131.39270	unknown	-
	−1	C_3_	^2^A	−2132.45687	−2131.65855	-	e
		C_1_	^2^A	−2132.47930	−2131.67406	0.610	-
**BEA**	0	C_3_	^1^A	−2589.67257	−2588.79282	0.000	-
	+1	C_1_	^2^A	−2589.38017	−2588.50454	unknown	-
	−1	C_1_	^2^A	−2589.62864	−2588.76092	unknown	-

**Table 4 molecules-28-06264-t004:** Molecular point groups, Γ, neighboring carbonyl O–O distances, *d*(O–O)_C=O_, bridging O–O distances, *d*(O–O)_bridge_, N–N distances, *d*(N–N), neighboring carbonyl O–O–O angles, *Θ*(O–O–O)_C=O_, bridging O–O–O angles, *Θ*(O–O–O)_bridge_, and N–N–N angles, *Θ*(N–N–N) of DFT optimized structures of enniatin B and beauvericin in various charge states *q*. All distances are in Å, all angles are in degrees.

Compound	*q*	Γ	*d*(O–O)_C=O_	*d*(O–O)_bridge_	*d*(N–N)	*Θ*(O–O–O)_C=O_	*Θ*(O–O–O)_bridge_	*Θ*(N–N–N)
**ENB**	0	C_3_	4.062 (3×)	6.037 (3×)	5.721 (3×)	67.3 (3×)	60.0 (3×)	60.0 (3×)
			3.845 (3×)			65.7 (3×)		
	+1	C_1_	3.380, 3.205	6.024	6.114	68.1, 65.9	63.4	57.7
			3.234, 3.009	6.393	5.775	80.4, 69.3	57.7	58.8
			3.742, 3.122	6.533	5.846	56.7, 64.4	61.1	63.5
	−1	C_3_	5.026 (3×)	5.389 (3×)	5.224 (3×)	78.0 (3×)	60.0 (3×)	60.0 (3×)
			4.943 (3×)			79.0 (3×)		
		C_1_	5.040, 5.027	5.261	5.184	76.1, 80.4	66.7	59.3
			4.983, 5.065	4.903	5.017	88.8, 78.8	54.5	62.6
			5.140, 5.043	4.665	4.954	82.7, 84.8	58.8	58.1
**BEA**	0	C_3_	3.194 (3×)	6.357 (3×)	5.920 (3×)	73.8 (3×)	60.0 (3×)	60.0 (3×)
			3.593 (3×)			60.9 (3×)		
	+1	C_1_	3.122, 3.380	6.392	6.114	69.3, 80.4	63.4	57.7
			3.205, 3.234	6.533	5.775	65.9, 68.1	61.1	63.5
			3.009, 3.742	6.024	5.846	64.4, 56.7	55.5	58.8
	−1	C_1_	3.211, 3.641	6.380	5.877	62.6, 73.0	60.0	59.9
			3.192, 3.607	6.409	5.878	61.8, 73.1	60.4	59.9
			3.184, 3.608	6.360	5.899	61.4, 71.0	59.6	60.2

**Table 5 molecules-28-06264-t005:** Molecular point groups, Γ, natural charges of carbonyl oxygens, *Q* (O)_C=O_, bridging oxygens, *Q*(O)_bridge_, and nitrogens, *Q*(N), natural spin populations of carbonyl oxygens, ρ(O)_C=O_, bridging oxygens, ρ(O)_bridge_, and nitrogens, ρ(N) of DFT optimized structures of enniatin B and beauvericin in various charged states *q*.

Compound	*q*	Γ	*Q*(O)_C=O_	*Q*(O)_bridge_	*Q*(N)	ρ(O)_C=O_	ρ(O)_bridge_	ρ(N)
**ENB**	0	C_3_	−0.639 (3×)	−0.585 (3×)	−0.554 (3×)	-	-	-
			−0.632 (3×)					
	+1	C_1_	−0.580, −0.454	−0.554	−0.232	0.078, 0.116	0.014	0.657
			−0.647, −0.623	−0.576	−0.566	0.005, 0.001	0.000	0.000
			−0.639, −0.617	−0.583	−0.551	0.000, 0.001	0.000	0.001
	−1	C_3_	−0.704 (3×)	−0.594 (3×)	−0.544 (3×)	0.068 (3×)	0.006 (3×)	0.003 (3×)
			−0.681 (3×)			0.029 (3×)		
		C_1_	−0.634, −0.684	−0.591	−0.549	0.000 (4×) 0.265, 0.003	0.000	0.001
			−0.824, −0.688	−0.669	−0.530		0.031	0.036
			−0.642, −0.651	−0.579	−0.546		0.000	0.000
**BEA**	0	C_3_	−0.607 (3×)	−0.598 (3×)	−0.560 (3×)	-	-	-
			−0.638 (3×)					
	+1	C_1_	−0.560, −0.665	−0.580	−0.213	0.040, 0.003	0.019	0.665 0.000 (2×)
			−0.612, −0.641	−0.593	−0.532	0.000 (2×)	0.000	
			−0.628, −0.442	−0.584	−0.559	−0.002, 0.174	−0.001	
	−1	C_1_	−0.619, −0.632	−0.600 (2×)	−0.559	0.007, 0.003	0.000 (2×) 0.001	0.001 0.000 (2×)
			−0.616, −0.631	−0.601	−0.560	0.004, 0.001		
			−0.624, −0.634		−0.601	0.012, 0.005		

## Data Availability

Data is contained within the article or Appendix A.

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
