# Peer review of "On the Potential Role of the (Pseudo-) Jahn–Teller Effect in the Membrane Transport Processes: Enniatin B and Beauvericin"

_molecules, 2023, doi:10.3390/molecules28176264_

Round 1

Reviewer 1 Report

1. In Abstract: abbreviations (JT) and (PJT) is not necessary to introduce here.

2. Abbreviations (JT) and (PJT) are introduced in line 60, once more in lines 75 and 104, at the same time "Jahn-Teller" in full presentation is given in lines 89 and 225.

3. In section 4 the authors  presented the methods and programs used for calculations. I suggest to explain why these tools were used in comparison with other possible.

4. The paper will be better if the authors indicate to what molecules the reported approach can be used also.

Author Response

Question: 1. In Abstract: abbreviations (JT) and (PJT) is not necessary to introduce here.

Answer: Deleted.

Question: 2. Abbreviations (JT) and (PJT) are introduced in line 60, once more in lines 75 and 104, at the same time "Jahn-Teller" in full presentation is given in lines 89 and 225.

Answer: Corrected.

Question: 3. In section 4 the authors  presented the methods and programs used for calculations. I suggest to explain why these tools were used in comparison with other possible.

Answer: Amended.

Question: 4. The paper will be better if the authors indicate to what molecules the reported approach can be used also.

Answer: Amended.

Reviewer 2 Report

In this work, the molecular structures of mycotoxins enniatin B and beauvericin were studied using density functional theory in various symmetry groups and singly charged states. The results show that the charge addition or removal causes significant structural changes, exhibiting Jahn-Teller symmetry descent. A detailed introduction is presented, interpreting the significance of this work. Theoretical methodologies are rationally provided, followed by clear results and discussion, leading to solid conclusions. I therefore recommend acceptance of this manuscript for publication in Molecules.

Here are my comments for this manuscript.

1. Please present the formula for Gibbs free energy calculation.

2. Line 154-155. The energies of anions are discussed here. However, I didn’t find these energies in Table 3. The authors should provide this information.

3. An error in Line 231.

Author Response

Question: 1. Please present the formula for Gibbs free energy calculation.

Answer: Presented in Supplementary Materials.

Question: 2. Line 154-155. The energies of anions are discussed here. However, I didn’t find these energies in Table 3. The authors should provide this information.

Answer: In Table 3, the energies of anions are presented at the lines corresponding to charges q = -1 (2nd column).

Question: 3. An error in Line 231.

Answer: We cannot find any error at this line.